# A Case-Control Study of the *APELA* Gene and Hypertensive Disorders of Pregnancy

**DOI:** 10.3390/medicina58050591

**Published:** 2022-04-26

**Authors:** Naomi Shimada, Tomohiro Nakayama, Hiroshi Umemura, Kei Kawana, Tatsuo Yamamoto, Seisaku Uchigasaki

**Affiliations:** 1Division of Legal Medicine, Department of Social Medicine, Nihon University School of Medicine, 30-1 Ooyaguchi-kamicho, Itabashi-ku, Tokyo 173-8610, Japan; shimada.naomi517@gmail.com (N.S.); uchigasaki.seisaku@nihon-u.ac.jp (S.U.); 2Division of Laboratory Medicine, Department of Pathology and Microbiology, Nihon University School of Medicine, 30-1 Ooyaguchi-kamicho, Itabashi-ku, Tokyo 173-8610, Japan; umemura.hiroshi@nihon-u.ac.jp; 3Division of Companion Diagnostics, Department of Pathology of Microbiology, Nihon University School of Medicine, 30-1 Ooyaguchi-kamicho, Itabashi-ku, Tokyo 173-8610, Japan; 4Department of Obstetrics and Gynecology, Nihon University School of Medicine, 30-1 Ooyaguchi-kamicho, Itabashi-ku, Tokyo 173-8610, Japan; kawana.kei@nihon-u.ac.jp (K.K.); yamamoto.tatsuo@nihon-u.ac.jp (T.Y.)

**Keywords:** hypertensive disorders of pregnancy, case-control study, variant, *APELA*, haplotype

## Abstract

Hypertensive disorders of pregnancy (HDPs) are believed to comprise a group of multifactorial genetic diseases. Recently, it was reported that *APELA*-knockout mice exhibited HDP-like symptoms, including proteinuria and elevated blood pressure due to defective placental angiogenesis. The aim of the present study is to determine the associations between HDPs and single-nucleotide variants or haplotypes in the human *APELA* gene through a case-control study. The subjects were 196 pregnant women with HDPs and a control group of 254 women without HDPs. Six single-nucleotide variants (rs2068792, rs13120303, rs4541465, rs13152225, rs78639146, and rs67448487) were selected from the *APELA* gene region. Although there were no significant differences for each single-nucleotide polymorphism in the case-control study, the frequency of the T-A haplotypes rs4541465–rs67448487 was significantly higher in the HDP group, especially in those with gestational hypertension, than in the control group. The results suggest that the *APELA* gene may be a disease-susceptibility gene for HDP.

## 1. Introduction

Maternal mortality, i.e., death due to preventable complications related to pregnancy or childbirth, is an international health care problem that causes the death of approximately 810 women every day. The aim of The Global Strategy for Women’s, Children’s and Adolescents’ Health of the World Health Organization (Sustainable Development Goal 3: Ensure healthy lives and promote wellbeing for all at all ages; https://www.who.int/sdg/targets/en/ 10 March 2022) is to reduce global maternal mortality to less than 70 per 100,000 live births by 2030. Hypertensive disorders of pregnancy (HDPs) are common: they occur in 3% to 5% of pregnant women (over 10,000 cases per year) in Japan, and about 7.5% of pregnancies in the world [1]. In Japan, the definition and classifications of HDPs were changed in 2018, and chronic hypertension (CH) in pregnancy was newly included in HDPs. As a result, HDPs are currently defined as hypertension during pregnancy. Many risk factors for HDPs have been reported, such as high body mass index, renal diseases, elderly pregnancy [2], and periodontal disease [3]. Thus, the reported incidence of HDPs is increasing due to an increase in the number of pregnant women with chronic health conditions, such as hypertension, obesity, and diabetes mellitus [2]. The incidence of HDPs in Japan is also thought to be increasing due to delayed childbearing in recent years, since the frequency of these lifestyle diseases tend to increase with increasing age. Pre-eclampsia (PE) is a disorder that occurs only during pregnancy and the postpartum period, and it is the leading cause of fetal and maternal morbidity and mortality. In addition, women with a history of PE have an increased risk of developing cardiovascular events later in life. Although PE is the cause of maternal mortality in at least 12% of all maternal mortality cases, its molecular etiology remains unclear. HDPs are believed to comprise a group of multifactorial genetic diseases [4,5]. In our past study, we reported the associations between HDPs and single-nucleotide variants (SNVs) or haplotypes in the human stromal interaction molecule 1 (*STIM1*) gene that regulates the concentration of Ca^2+^ and vascular contraction [6]. Furthermore, in a haplotype-based case-control study of the *PPP1R12A* gene, a significant difference was found between the control group and the HDP group [7].

The peptide hormone APELA (apelin receptor, also known as Toddler, ELA, or ELABELA) was discovered in 2013. It plays an essential role in the embryonic cardiovascular system formation in zebrafish through the apelin receptor (APJ) [8,9]. The etymology of apelin is thought to be derived from “an adipokine with pleiotropic effects in many physiological processes of the body” [10]. APELA is an endogenous ligand of APJ, and it competes with apelin. APELA is secreted as a circulating hormone that works in paracrine and endocrine manners, and it is expressed by a few organs, including the placenta and kidney [10,11]. It was found that *APELA*-knockout mice exhibited PE-like symptoms, including proteinuria and elevated blood pressure due to defective placental angiogenesis [12].

The aim of the present study is to determine the associations between HDPs and SNVs or haplotypes in the human *APELA* gene through a case-control study.

## 2. Materials and Methods

### 2.1. Subjects

The present study group involved 196 pregnant Japanese women with HDPs, and 254 Japanese women without HDPs as the control group. The target sample size was determined using the following criteria: the case group needed to be 100 or more, and the number of subjects in the control group should be about 200, because it is better to have twice the number of subjects in the control group than in the case group [13]. All participants were recruited from among the patients and healthy volunteers who visited the Department of Obstetrics and Gynecology, Nihon University School of Medicine Itabashi Hospital, Tokyo, Japan, between 2006 and 2015. For data analysis in comparison to the control group, the HDP group was divided by disease type into three groups: those with gestational hypertension (GH), PE, or superimposed PE (SPE). HDPs were defined as systolic blood pressure ≥ 140 mmHg or diastolic blood pressure ≥ 90 mmHg that developed after 20 weeks of gestation and improved by 12 weeks’ postpartum. GH was defined as hypertension without proteinuria that appeared after 20 weeks’ gestation and normalized by 12 weeks’ post-delivery. PE was defined as hypertension that appeared after 20 weeks gestation with the presence of proteinuria and normalized by 12 weeks’ post-delivery. SPE was defined as hypertension that appeared after 20 weeks’ gestation if kidney disease with proteinuria as the only symptom was present before pregnancy or before 20 weeks’ gestation, hypertension that worsens after 20 weeks’ gestation if both hypertension and proteinuria were previously present before pregnancy or before 20 weeks’ gestation, or hypertension that was present before pregnancy or before 20 weeks’ gestation, together with proteinuria after 20 weeks’ gestation. Hypertension was defined as systolic blood pressure ≥ 140 mmHg or diastolic blood pressure ≥ 90 mmHg. Proteinuria was defined as the detection of ≥300 mg/day of protein in 24 h urine. Women who did not meet the diagnostic criteria for HDPs were enrolled in the control group. The control group included healthy primigravidas who had no evidence of any medical or obstetrical complications. Diagnoses of HDPs and subtypes were performed based on the observable clinical course, and in principle, clinical data were obtained during pregnancy.

The protocol of this study was approved by the Ethics Committee of Nihon University School of Medicine (Approval No. 249-1). Written, informed consent was obtained from each participant.

### 2.2. Genotyping

The *APELA* gene is located on chromosome 4q32.3, and it contains four exons (Figure 1).

Six SNVs in the human *APELA* gene were selected as genetic markers for the association experiment. All six SNVs were selected from the NCBI website (accession numbers rs2068792, rs13120303, rs4541465, rs13152225, rs78639146, and rs67448487) as SNVs with a minor allele frequency (MAF) of over 2% in the Japanese population. Four of the SNVs are non-neighboring: rs13120303 and rs13152225 are located in introns; rs4541465 is a synonymous SNV that does not result in a change of amino acids; rs2068792 is located in the 5′-UTR; and rs78639146 and rs67448487 are located in the 3′-UTR.

The MAF and the nucleotide context sequence [VIC/FAM] around each SNV were as follows: rs2068792, MAF (C) = 0.05, 5′-TAAGTCATTCTCTTCTTGGAGCTTT[C/T]GGAGTACACTTCCACTAAAGTTATA-3′; rs13120303, MAF (G) = 0.43, 5′-GATATTACTATTAAGATGTTGCTAT[A/G]TTGCTCAACGATGGAGTTAAAAAAA-3′; rs4541465, MAF (T) = 0.41, 5′-TGCCTCTCCATTCACGAGTACCCTT[C/T]CCCTGAGGTATTTCTGACAGAAAAT-3′; rs13152225, MAF (A) = 0.14, 5′-AAACTATGACATTATTAGTATATTC[A/G]CTTCAGCTCTCTTATACTTGCTGCT-3′; rs78639146, MAF (A) = 0.02, 5′-ACATTTGAACTCCATTTTTGAAAAA[A/T]ATAAAAACTAACACCCACGAAAAAT-3′; and rs67448487, MAF (A) = 0.15, 5′-TAATCCCAGCACTTTGGGAATCTAA[G/A]ACAGGAGGATTGCTTGAAGCAAAGA-3′.

All genomic DNA was obtained from peripheral blood leukocytes using the phenol and chloroform extraction method. The concentration of nucleic acids was adjusted to 100 ng/μL with a spectrophotometer. TaqMan PCR and the TaqMan SNP Genotyping Assay (Applied Biosystems, Foster City, CA, USA) were used to determine the genotypes according to the manufacturer’s instructions. The extracted DNA, TaqMan Universal PCR Master Mix, Assay Mix (TaqMan probe and primer), and distilled water were mixed and injected into a 96-well plate for reaction. For DNA amplification, DNA was placed in a 2720 Thermal Cycler (Applied Biosystems), which was set to 95 °C for 10 min, then 50 to 70 cycles at 92 °C for 15 s, 60 °C for 1 min, and finally held at 4 °C. An ABI PRISM 7700 Sequence Detector (PE Biosystems, Rotkreuz, Switzerland) and 7500 Fast Real-Time PCR System and Detector v.1.7 (Applied Biosystems) were used to detect the TaqMan PCR fluorescent dye and analyze the obtained data.

### 2.3. Statistical Analysis

The analyses of rs2068792 and rs78639146 were omitted, because no variants were found among the participants in genotype and haplotype. The clinical data of all continuous variables are shown as means ± standard deviation (SD). Differences in continuous variables between the HDP and control groups were analyzed with the Mann–Whitney U-test. Using Fisher’s exact test, the distribution of genotypes between the HDP and control groups was analyzed. A linkage disequilibrium analysis and haplotype-based case-control study were performed using SNPAlyze version 9 software (Dynacom, Chiba, Japan). Haplotype blocks were defined as |D’| values of >0.5 to assign all four SNV locations to the same block (data not shown). SNVs with an r^2^ value of <0.5 were selected as tagged. The r^2^ values between rs13120303 and rs4541465 and between rs13152225 and rs67448487 were >0.5 (Table 1), and SNVs with a higher MAF were preferred in the haplotype-based case-control study.

In the haplotype-based case-control study, haplotypes with frequencies of less than 0.02 were excluded. The frequency distribution was examined by the *χ^2^*-test. Significance was established at *p* < 0.05. Statistical analyses were performed using SPSS^®^ software for Windows, version 25 (SPSS, Chicago, IL, USA).

## 3. Results

The clinical characteristics of the study participants are shown in Table 2.

The prevalence of having a family history of hypertension was higher in the HDP, PE, and SPE groups than in the control group. SBP, DBP, and the body mass index (BMI) before pregnancy were higher, and the gestational age at delivery and the birth weight of the neonates were lower in the HDP, GH, PE, and SPE groups than in the control group. The BMI at delivery was higher in the HDP, GH, and PE groups than in the control group. The Apgar score at 5 min was lower in the HDP, PE, and SPE groups than in the control group.

The genotypes, dominant models, recessive models, and alleles are shown in Table 3.

The genotype frequency distribution of the four SNVs in the control group conformed to the Hardy–Weinberg equilibrium with *p*-values > 0.05 (rs13120303: *p* = 0.06; rs4541465: *p* = 0.24; rs13152225: *p* = 0.22; and rs67448487: *p* = 0.17).

Among the SNVs analyzed, there were no significant differences in the allele or genotype frequency distribution between the HDP and control groups. There were no significant differences in the genotypes and alleles among the GH, PE, and SPE groups.

The haplotype-based control study results are shown in Table 4.

The T-A haplotypes rs4541465–rs67448487 were significantly more common in the GH group than in the control group, with a permutation *p*-value of 2%. The T-A haplotypes rs4541465–rs13152225 were also significantly more common in the HDP and GH groups than in the control group. The subjects with this particular haplotype have not been shown to have specific clinical data (data not shown).

## 4. Discussion

HDPs are known to occur in approximately 1 in 20 pregnant women, and if they occur within 4 weeks of gestation, they can become severe, and caution is required. HDPs are considered to comprise a group of multifactorial genetic diseases, and although association studies of HDPs have been conducted in recent years, no specific causative gene has yet been identified [14,15]. Hypertension, a history of HDPs, a family history of hypertension, obesity, an advanced maternal age pregnancy, diabetes mellitus, multiple pregnancies, primigravida, and renal disease are known risk factors for HDPs. In the present study, there were significant differences between the HDP and control groups in the BMI at pregnancy and the numbers of individuals with hypertension (7.78% vs. 1.45%), a family history of hypertension (35.9% vs. 23.1%), or a history of HDPs (31.0% vs. 0.0%), but the numbers of individuals with diabetes mellitus (0.0% vs. 0.48%) or renal disease (2.29% vs. 0.97%), the proportion of primigravidas (58.2% vs. 63.1%), and the body weight gained during pregnancy (9.4 kg vs. 10.2 kg) did not differ significantly between the groups. The age at delivery did not differ significantly between the HDP and control groups (31.8 ± 6.3 years vs. 31.2 ± 6.7 years), but there was a significant difference between the SPE and control groups (35.3 ± 6.1 years vs. 31.2 ± 6.7 years). When comparing the HDP and control groups, there were significant differences in the gestational age at delivery (35.1 ± 4.0 weeks vs. 38.7 ± 1.7 weeks), birth weight of the neonates (2150.8 ± 863.9 g vs. 3042.5 ± 479.7 g), and the Apgar score at 5 min (7.7 ± 2.3 vs. 8.7 ± 0.8), but not in the body weight gained during pregnancy (9.4 ± 6.2 kg vs. 10.2 ± 4.2 kg). Since there is no safe and effective treatment for HDPs, and the HDPs resolve after delivery, a cesarean section is often performed before full-term pregnancy (37 to 42 weeks of gestation), and the proportion of infants with a low birth weight is increasing.

Apelin, extracted from the bovine stomach as an endogenous ligand for APJ, was first identified in 1998 by Tatemoto et al. [16]. The apelin/APJ system regulates body fluid homeostasis, expands peripheral blood vessels, and enhances myocardial contractility [17,18]. It has been reported that apelin and APELA both act via the APJ, and they have similar functions [9]. However, little is known about the biological properties and functions of APELA. Key aspects of maternal cardiovascular homeostasis and placental development are regulated by the APELA signaling mechanism. APELA plays a role in nitric oxide (NO) production to increase uterine blood flow, promote cytotrophoblast invasion and angiogenesis, and reduce apoptosis due to oxidative stress in the placenta [19]. APELA also has a paracrine signaling role in promoting trophoblast differentiation into invasive extravillous trophoblasts and invasion for subsequent spiral artery remodeling [12]. Therefore, a deficiency of APELA might cause PE-like symptoms, including impaired placental vascular functions. A placenta suffering from PE is characterized by endothelial dysfunction and poor trophoblastic invasion.

Many researchers are now reporting about APELA. Deniz et al. reported that the APELA amounts and NO levels in maternal blood were significantly lower in the PE group, especially in women with severe PE, than in control groups [20]. However, Pritchard et al. reported that APELA levels were not decreased in the maternal circulation or placenta in subjects with PE [21]. Another group reported that the levels of mRNA, protein expression, and APELA were significantly decreased in late-onset PE due to the effect of APELA on the migration and proliferation of human trophoblast cells [22]. The infusion of APELA had vasodilator activity, and it has been shown to decrease blood pressure in rat experiments. Although the intracellular signal transduction mechanism has not yet been clarified, it has been suggested that APELA may have an antihypertensive effect through the inhibition of ERK activation in an NO-independent manner [12]. APELA may be a potential candidate for the treatment of renal ischemia-reperfusion injury and acute kidney injury [23]. APELA deficiency may lead to placental ischemia and maternal kidney/endothelial dysfunction, and contribute to either GH or PE in pregnant women [12,24]. Apelin has been confirmed to be strongly associated with the occurrence and development of hypertension, and apelin and APELA share the APJ and exhibit a similar cardiovascular profile [25,26]. A previous study reported that the GH group had a higher plasma APELA concentration than the control group [27]. In a preliminary study performed in 2019, Li et al. demonstrated that circulating APELA levels were lower in patients with essential hypertension due to impaired vascular function [28].

The present study showed that the haplotype of the *APELA* gene is related to HDPs, especially GH. To determine the region of the strongest association with a disease, it is necessary to conduct an association study of the nearby polymorphisms. For such an investigation, haplotype-based case-control studies may be more advantageous than an analysis based on individual SNVs, especially for identifying susceptibility markers for multifactorial disorders [29]. Haplotypes are a sequence of alleles on chromosomes derived from one parent, and they are more diverse than individual SNVs. Therefore, the detection sensitivity of disease-susceptibility genes can be increased by adding an association analysis using haplotypes. In addition, the inclusion of an association analysis using haplotypes would greatly reduce the time and effort needed for the analysis and enable the efficient determination of the presence or absence of disease susceptibility. Indeed, we identified significant genetic markers using individual SNVs, even though there was no significant difference in the overall distribution of the allele or genotype frequencies of SNVs between the patient and control groups [30]. The significant T-A haplotypes, i.e., rs4541465, which is a synonymous SNV, and rs67448487, which is located in the 3′-untranslated region (UTR), do not directly affect changes in amino acid sequences or the construction of the *APELA* gene. There were variants linked to the haplotypes that were frequently observed in HDP in the present study, and it is possible that these variants are related to the onset of disease by reducing the function and expression level of the *APELA* gene.

Yang et al. reported that rare variants in the 5′-UTR of the *APELA* gene were associated with PE, but there have been no studies to date indicating a relationship between the *APELA* gene and HDPs through a haplotype-based case-control study [31]. The HDP-sensitive haplotype identified in the present study might affect variants in *APELA* that cause a genetic disorder. To definitively prove this, it will be necessary to survey the whole exon region of participants who have a susceptibility haplotype of the *APELA* gene via nucleotide sequencing. Although the subjects with this particular haplotype have not been shown to have specific clinical data, it will be necessary to focus on the promoter region and translated region to search for variants that may be linked to the haplotypes found in the present study in the future.

In the present study, strict criteria were applied for the control group by selecting only healthy primigravidas who had no evidence of any medical or obstetrical complications. These strict criteria for the control group ensured increased sample confidence for our study data. Although the sample size of this study was not large, the number of subjects in the disease group needed to be at least 100, and the number of subjects in the control group should be about 200, because it is better to have twice the number of subjects in the control group than in the disease group for case-control studies [13]. By increasing the number of subjects in the future, the associations of genetic factors will become clearer.

## 5. Conclusions

The frequency of the T-A haplotypes rs4541465–rs67448487 was significantly higher in the HDP group, especially in those with gestational hypertension, than in the control group. These results suggest that the *APELA* gene may be a disease-susceptibility gene for HDPs. If these findings are confirmed, further research is expected to identify the relationship between the underlying biological mechanisms of HDPs and the functional variants of these genes.

## Figures and Tables

**Figure 1 medicina-58-00591-f001:**
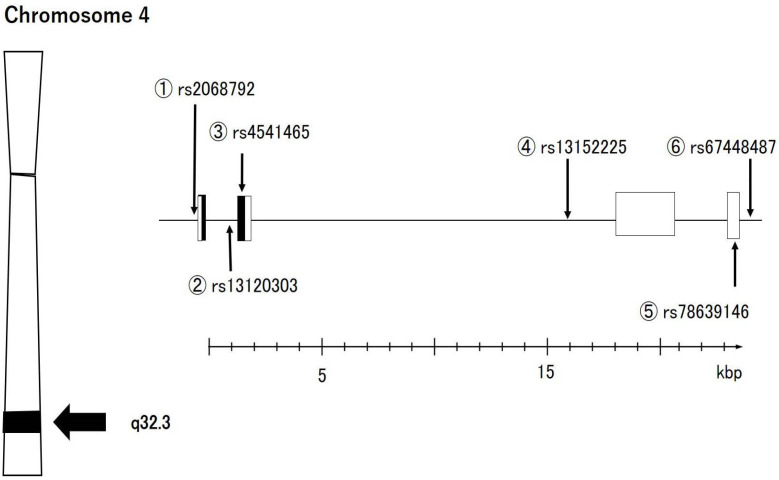
Organization of the *APELA* gene and location of SNVs. The *APELA* gene is located on 4q32.3. The arrows indicate the locations of the SNVs. Open and closed boxes indicate untranslated regions and translated regions in exons, respectively. Lines indicate introns.

**Table 1 medicina-58-00591-t001:** Linkage disequilibrium patterns (r^2^).

	rs13120303	rs4541465	rs13152225	rs67448487
rs13120303		0.69	9.6 × 10^−5^	3.7 × 10^−3^
rs4541465			0.058	0.047
rs13152225				0.71
rs67448487				

**Table 2 medicina-58-00591-t002:** Characteristics of the study participants.

	Control	HDPs		GH		PE		SPE	
	(*n* = 254)	(*n* = 196)		(*n* = 78)		(*n* = 106)		(*n* = 12)	
			*p*-Values		*p*-Values		*p*-Values		*p*-Values
Age at delivery (years)	31.2 ± 6.7	31.8 ± 6.3	0.388	32.1 ± 6.2	0.354	31.2 ± 6.3	0.985	35.3 ± 6.1	0.036 *
Proportion of primigravidae (%)	63.1	58.2	0.387	54.1	0.312	59.8	0.608	58.3	0.744
Family history of hypertension (%)	23.1	35.9	0.01 *	30.2	0.324	36.4	0.02 *	54.5	0.019 *
History of HDPs (%)	0.0	31.0	<0.001 *	30.4	<0.001 *	32.7	<0.001 *	22.2	<0.001 *
Hypertension (%)	1.45	7.78	0.003 *	0.0	0.392	1.89	0.77	91.7	<0.001 *
Diabetes mellitus (%)	0.48	0.0	0.368	0.0	0.622	0.0	0.474	0.0	0.809
Renal disease (%)	0.97	2.99	0.15	2	0.541	3.77	0.087	0.0	0.732
Autoimmune disease (%)	1.45	0.0	0.118	0.0	0.392	0.0	0.213	0.0	0.675
Systolic blood pressure (mmHg)	118.4 ± 17.5	163.8 ± 22.8	<0.001 *	148.8 ± 18.2	<0.001 *	168.9 ± 21.2	<0.001 *	183.1 ± 20.4	<0.001 *
Diastolic blood pressure (mmHg)	73.4 ± 11.7	98.8 ± 17.4	<0.001 *	90.3 ± 16.3	<0.001 *	101.2 ± 16.8	<0.001 *	112.7 ± 11.3	<0.001 *
BMI before pregnancy (kg/m^2^)	20.7 ± 3.0	23.2 ± 4.8	<0.001 *	22.8 ± 3.9	0.001 *	23.0 ± 4.7	<0.001 *	26.5 ± 7.4	0.019 *
BMI at delivery (kg/m^2^)	24.8 ± 2.7	26.9 ± 4.3	<0.001 *	26.8 ± 3.8	0.002 *	26.6 ± 4.1	<0.001 *	29.3 ± 7.2	0.052
Body weight gained during pregnancy (kg)	10.2 ± 4.2	9.4 ± 6.2	0.231	9.8 ± 6.0	0.694	9.6 ± 6.5	0.423	7.1 ± 5.3	0.019 *
Gestational age at delivery (weeks)	38.7 ± 1.7	35.1 ± 4.0	<0.001 *	37.0 ± 2.7	<0.001 *	34.4 ± 4.1	<0.001 *	33.0 ± 5.5	0.004 *
Birth weight of the neonate (g)	3042.5 ± 479.7	2150.8 ± 863.9	<0.001 *	2565.5 ± 692.6	<0.001 *	1997.8 ± 833.1	<0.001 *	1835.9 ± 1156.0	0.003 *
Apgar score at 5 min	8.7 ± 0.8	7.7 ± 2.3	<0.001 *	8.5 ± 1.0	0.408	7.4 ± 2.5	<0.001 *	6.5 ± 3.0	0.027 *

Values of the continuous variables are expressed as the means ± standard deviation. HDPs: hypertensive disorders of pregnancy; GH: gestational hypertension; PE: pre-eclampsia; SPE: superimposed pre-eclampsia; BMI: body mass index. * *p* < 0.05.

**Table 3 medicina-58-00591-t003:** Genotype and allele distributions among the control, HDP, GH, PE, and SPE groups.

				Control	HDP		GH		PE		SPE	
				(*n* = 254)	(*n* = 196)		(*n* = 78)		(*n* = 106)		(*n* = 12)	
Variants						*p*-Values		*p*-Values		*p*-Values		*p*-Values
rs13120303	Genotype		AA	45 (0.177)	31 (0.158)	0.111	12 (0.154)	0.162	17 (0.160)	0.265	2 (0.167)	0.995
			AG	106 (0.417)	101 (0.515)		42 (0.538)		54 (0.509)		5 (0.417)	
			GG	103 (0.406)	64 (0.327)		24 (0.308)		35 (0.330)		5 (0.417)	
		Dominant model	AA	45 (0.177)	31 (0.158)	0.594	12 (0.154)	0.633	17 (0.160)	0.701	2 (0.167)	0.926
			GG + AG	209 (0.823)	165 (0.842)		66 (0.846)		89 (0.840)		10 (0.833)	
		Recessive model	GG	103 (0.406)	64 (0.327)	0.086	24 (0.308)	0.120	35 (0.330)	0.180	5 (0.417)	0.939
			AG + AA	151 (0.594)	132 (0.673)		54 (0.692)		71 (0.670)		7 (0.583)	
	Allele		A	196 (0.386)	163 (0.416)	0.362	66 (0.423)	0.405	88 (0.415)	0.464	9 (0.375)	0.915
			G	312 (0.614)	229 (0.584)		90 (0.577)		124 (0.585)		15 (0.625)	
rs4541465	Genotype		CC	59 (0.232)	42 (0.214)	0.375	14 (0.179)	0.199	24 (0.226)	0.652	4 (0.333)	0.629
			CT	117 (0.461)	103 (0.526)		45 (0.577)		54 (0.509)		4 (0.333)	
			TT	78 (0.307)	51 (0.260)		19 (0.244)		28 (0.264)		4 (0.333)	
		Dominant model	CC	59 (0.232)	42 (0.214)	0.650	14 (0.179)	0.325	24 (0.226)	0.904	4 (0.333)	0.421
			TT + CT	195 (0.768)	154 (0.786)		64 (0.821)		82 (0.774)		8 (0.667)	
		Recessive model	TT	78 (0.307)	51 (0.260)	0.276	19 (0.244)	0.281	28 (0.264)	0.415	4 (0.333)	0.847
			CT + CC	176 (0.693)	145 (0.740)		59 (0.756)		78 (0.736)		8 (0.667)	
	Allele		C	235 (0.463)	187 (0.477)	0.667	73 (0.468)	0.907	102 (0.481)	0.650	12 (0.500)	0.720
			T	273 (0.537)	205 (0.523)		83 (0.532)		110 (0.519)		12 (0.500)	
rs13152225	Genotype		AA	8 (0.031)	3 (0.015)	0.072	2 (0.026)	0.201	1 (0.009)	0.164	0 (0.000)	0.623
			AG	59 (0.232)	63 (0.321)		26 (0.333)		33 (0.311)		4 (0.333)	
			GG	187 (0.736)	130 (0.663)		50 (0.641)		72 (0.679)		8 (0.667)	
		Dominant model	AA	8 (0.031)	3 (0.015)	0.270	2 (0.026)	0.791	1 (0.009)	0.222	0 (0.000)	0.533
			GG + AG	246 (0.969)	193 (0.985)		76 (0.974)		105 (0.991)		12 (1.000)	
		Recessive model	GG	187 (0.736)	130 (0.663)	0.093	50 (0.641)	0.104	72 (0.679)	0.273	8 (0.667)	0.595
			AG + AA	67 (0.264)	66 (0.337)		28 (0.359)		34 (0.321)		4 (0.333)	
	Allele		A	75 (0.148)	69 (0.176)	0.250	30 (0.192)	0.181	35 (0.165)	0.553	4 (0.167)	0.798
			G	433 (0.852)	323 (0.824)		126 (0.808)		177 (0.835)		20 (0.833)	
rs67448487	Genotype		GG	184 (0.724)	132 (0.673)	0.060	51 (0.654)	0.058	72 (0.679)	0.370	9 (0.750)	0.803
			GA	61 (0.240)	62 (0.316)		27 (0.346)		32 (0.302)		3 (0.250)	
			AA	9 (0.035)	2 (0.010)		0 (0.000)		2 (0.019)		0 (0.000)	
		Recessive model	GG	184 (0.724)	132 (0.673)	0.241	51 (0.654)	0.231	72 (0.679)	0.389	9 (0.750)	0.846
			GA + AA	70 (0.276)	64 (0.327)		27 (0.346)		34 (0.321)		3 (0.250)	
		Dominant model	AA	9 (0.035)	2 (0.010)	0.086	0 (0.000)	0.092	2 (0.063)	0.405	0 (0.000)	0.507
			GG + GA	245 (0.965)	194 (0.990)		78 (1.000)		104 (0.937)		12 (1.000)	
	Allele		G	429 (0.844)	326 (0.832)	0.603	129 (0.827)	0.600	176 (0.830)	0.633	21 (0.875)	0.686
			A	79 (0.156)	66 (0.168)		27 (0.173)		36 (0.170)		3 (0.125)	

The frequencies are shown in parentheses. *p*-values were determined in comparison to the control group by Fischer’s exact test. *p* values < 0.05 are considered to be statistically significant. HDP: hypertensive disorders of pregnancy; GH: gestational hypertension; PE: pre-eclampsia; SPE: superimposed pre-eclampsia.

**Table 4 medicina-58-00591-t004:** Haplotype-based case-control study between the control and HDP groups using single-nucleotide variants.

		Control	HDP		Control	GH		Control	PE		Control	SPE	
Haplotype		Frequency (%)		*p*-Values	Frequency (%)		*p*-Values	Frequency (%)		*p*-Values	Frequency (%)		*p*-Values
rs4541465	rs13152225												
T	G	50.07	45.14	0.156	50.07	45.41	0.363	50.07	45.03	0.198	50.07	44.61	0.86
C	G	35.17	37.25	0.582	35.17	35.36	1	35.17	38.46	0.402	35.17	38.72	0.813
C	A	11.09	10.46	0.821	11.09	11.44	0.898	11.09	9.66	0.598	11.09	11.28	0.739
T	A	3.67	7.15	0.045 *	3.67	7.8	0.046 *	3.67	6.85	0.077	3.67	5.39	0.606
rs4541465	rs67448487												
T	G	49.31	44.99	0.21	49.31	44.5	0.338	49.31	44.55	0.254	49.31	50	1
C	G	35.14	38.17	0.351	35.14	38.19	0.537	35.14	38.47	0.414	35.14	37.5	0.83
C	A	11.12	9.53	0.508	11.12	8.61	0.36	11.12	9.65	0.601	11.12	12.5	0.752
T	A	4.43	7.31	0.072	4.43	8.7	0.047 *	4.43	7.33	0.112	4.43	0.01	0.752

HDP: hypertensive disorders of pregnancy; GH: gestational hypertension; PE: pre-eclampsia; SPE: superimposed pre-eclampsia. * *p* < 0.05.

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
