# Peer review of "A Case-Control Study of the APELA Gene and Hypertensive Disorders of Pregnancy"

_medicina, 2022, doi:10.3390/medicina58050591_

Round 1
Reviewer 1 Report
In this study the authors examine a previously suggested role of the APELA gene in relation to hypertensive disorders of pregnancy in murine models.
There are very few details given regarding the study population and how this group was chosen. Did the authors perform a sample size calculation prior to the study? How and when were the women recruited? During pregnancy or after delivery? Any risk of bias?
Unfortunately, it is very difficult to read Table 3.
There is no mentioning of Table 4 in the text.
It would be interesting to see if any of the SNVs were associated with any of the characteristics of the participants. Any signs of pleiotropy?
The study by Pritchard et al (ELABELA/APELA Levels Are Not Decreased in
the Maternal Circulation or Placenta among Women with Preeclampsia) should also be cited.
Author Response
Point-by-point responses to reviewers’ comments
Manuscript ID: medicina-1653150
Article type: Article
Title: A Case-control Study of the APELA Gene and Hypertensive Disorders of Pregnancy
Thank you very much for your valuable comments on our manuscript. We have revised our manuscript according to the comments and suggestions. The changes are indicated below in our point-by-point responses to these items.
Comments and Suggestions from the reviewer:
Reviewer 1
In this study the authors examine a previously suggested role of the APELA gene in relation to hypertensive disorders of pregnancy in murine models.
There are very few details given regarding the study population and how this group was chosen. Did the authors perform a sample size calculation prior to the study?
Response
->
The target sample size is described with references in the Materials and Methods.
How and when were the women recruited?
Response
->
“All participants were recruited from among the patients and healthy volunteers who visited the Department of Obstetrics and Gynecology, Nihon University School of Medicine Itabashi Hospital, Tokyo, Japan, between 2006 and 2015.” This information is included in the Materials and Methods.
During pregnancy or after delivery? Any risk of bias?
Response
->
Diagnoses of HDPs and subtypes were performed by the observable clinical course, and in principle, clinical data were obtained during pregnancy. This is included in the Materials and Methods.
Unfortunately, it is very difficult to read Table 3.
Response
->
The font of Table 3 has been enlarged to make it easier to read.
There is no mentioning of Table 4 in the text.
Response
->
According to the reviewer’s comment, we have cited Table 4 and the relevant information in the text.
It would be interesting to see if any of the SNVs were associated with any of the characteristics of the participants. Any signs of pleiotropy?
Response
->
Subjects with this particular haplotype have not been shown to have specific clinical data. This was included in the Results and discussed in the Discussion.
The study by Pritchard et al (ELABELA/APELA Levels Are Not Decreased in the Maternal Circulation or Placenta among Women with Preeclampsia) should also be cited.
Response
->
We have added this reference in the Discussion, as suggested.
Reviewer 2 Report
It s interesting and well prepared manuscript which is worth publishing.
I do recommend only to explain APELA abbreviation because you do not write anywhere about it . Put it into the bracket. It must be known to the readers because it does not have to be obvious to everybody.
Author Response
Please see tha attachement.

Reviewer 3 Report
Review
Thank you for the opportunity to review your work. Below you can find few recommendations:
Line 48: it is mentioned that PHD is common, but it is pointed out only the situation in Japan; please present an worldwide overview.
Line 51: there are many more risk factors that must be mentioned
Line 89: superimposed PE means chronic hypertension associated with pregnancy? Please explain
Line 165-what is the difference between PHD and GH??? Please explain
Line 191-193: there are mentioned only the numbers of individuals. Please add proportions, are easier to follow
Line 196-198: all the results are not very clear presented. Please clarify
Line 200: “a cesarean section is often performed before the third trimester of pregnancy (37 to 42 weeks of gestation” – please correct and clarify
Conclusion: please adapt your conclusions to the results obtained in the study
Please add information about patient informed consent to take part in your study.
Author Response
Please see the attachment.

This manuscript is a resubmission of an earlier submission. The following is a list of the peer review reports and author responses from that submission.